# SPATIOTEMPORAL MODELING OF MULTIVARIATE SIGNALS WITH GRAPH NEURAL NETWORKS AND STRUCTURED STATE SPACE MODELS

## ABSTRACT

Multivariate signals are prevalent in various domains, such as healthcare, transportation systems, and space sciences. Modeling spatiotemporal dependencies in multivariate signals is challenging due to (1) long-range temporal dependencies and (2) complex spatial correlations between sensors. To address these challenges, we propose representing multivariate signals as graphs and introduce GRAPHS4MER, a general graph neural network (GNN) architecture that captures both spatial and temporal dependencies in multivariate signals. Specifically, (1) we leverage Structured State Spaces model (S4), a state-of-the-art sequence model, to capture long-term temporal dependencies and (2) we propose a graph structure learning layer in GRAPHS4MER to learn dynamically evolving graph structures in the data. We evaluate our proposed model on three distinct tasks and show that GRAPHS4MER consistently improves over existing models, including (1) seizure detection from electroencephalography signals, outperforming a previous GNN with self-supervised pretraining by 3.1 points in AUROC; (2) sleep staging from polysomnography signals, a 4.1 points improvement in macro-F1 score compared to existing sleep staging models; and (3) traffic forecasting, reducing MAE by 8.8% compared to existing GNNs and by 1.4% compared to Transformer-based models.

## 1 INTRODUCTION

Multivariate signals are time series data measured by multiple sensors and are prevalent in many real-world applications, including healthcare (Mincholé et al., 2019), transportation systems (Ermagun & Levinson, 2018), power systems (Negnevitsky et al., 2009), and space sciences (Camporeale et al., 2018). An example multivariate signal is scalp electroencephalograms (EEGs), which measure brain electrical activities using sensors placed on an individual's scalp.

Several challenges exist in modeling spatiotemporal dependencies in multivariate signals. First, many types of signals are sampled at a high sampling rate, which results in long sequences that can be up to tens of thousands of time steps. Moreover, multivariate signals often involve long-range temporal correlations (Berthouze et al., 2010). Prior studies on modeling long signals often preprocess the raw signals using frequency transformations (Tang et al., 2022b; Asif et al., 2020; Shoeibi et al., 2021; Covert et al., 2019; Guillot et al., 2020; Guillot & Thorey, 2021) or divide the signals into short windows and aggregate model predictions post-hoc (Phan & Mikkelsen, 2022; Pradhan et al., 2022). However, such preprocessing steps may discard important information encoded in raw signals, as well as neglect long-range temporal dependencies in the signals. Therefore, a model that is capable of modeling long-range temporal correlations in raw signals is needed.

Deep sequence models, including recurrent neural networks (RNNs), convolutional neural networks (CNNs), and Transformers, have specialized variants for handling long sequences (Arjovsky et al., 2016; Erichson et al., 2021; Katharopoulos et al., 2020; Choromanski et al., 2021). However, they struggle to scale to long sequences of tens of thousands of time steps (Tay et al., 2020). Recently, the Structured State Space sequence model (S4) (Gu et al., 2022), a deep sequence model based on the classic state space model, has achieved state-of-the-art performance on challenging long

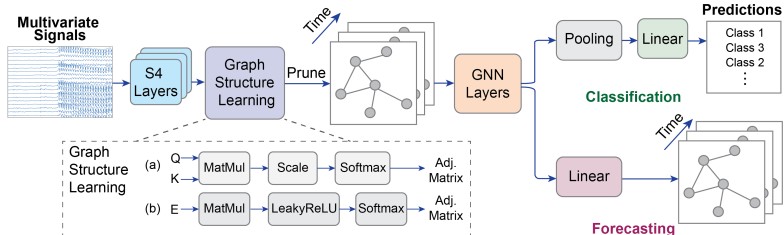

Figure 1: **Architecture of GRAPHS4MER**. The model has three main components: (1) stacked S4 layers to learn temporal dependencies in each sensor independently; (2) a graph structure learning (GSL) layer to learn dynamically evolving graph structures; (3) GNN layers to learn spatial dependencies based on S4 embeddings and learned graph structures. For GSL, we adopt (a) self-attention and (b) a learnable embedding for inductive and transductive settings, respectively.

sequence modeling tasks, such as the Long Range Arena benchmark (Tay et al., 2020), raw speech classification (Gu et al., 2022), and audio generation (Goel et al., 2022).

Second, sensors have complex, non-Euclidean spatial correlations. For example, EEG sensors measure highly correlated yet unique electrical activities from different brain regions (Michel & Murray, 2012); traffic speeds are correlated not only based on physical distances between traffic sensors, but also dependent on the traffic flows (Li et al., 2018b). Graphs are data structures that can model complex, non-Euclidean correlations in the data (Chami et al., 2022; Bronstein et al., 2017). Previous works have adopted temporal graph neural networks (GNNs) in modeling multivariate time series, such as EEG-based seizure detection (Covert et al., 2019) and classification (Tang et al., 2022b), traffic forecasting (Li et al., 2018b; Wu et al., 2019; Zheng et al., 2020b; Jiang & Luo, 2022; Tian & Chan, 2021), and pandemic forecasting (Panagopoulos et al., 2021; Kapoor et al., 2020). Nevertheless, most of these studies use sequences up to hundreds of time steps and require a predefined, static graph structure. However, the graph structure of multivariate signals may not be easily defined due to unknown sensor locations. For instance, while EEG sensors are typically placed according to the 10-20 standard placement (Jasper, 1958), the exact locations of sensors vary in each individual's recordings due to the variability in individual head size. Moreover, the underlying graph connectivity can evolve over time due to temporal dynamics in the data. Hence, when graph structures cannot be easily predefined, the ability to dynamically learn the underlying graph structures is highly desired.

Graph structure learning (GSL) aims to jointly learn an optimized graph structure and its node and graph representations (Zhu et al., 2021). GSL techniques have been used in non-temporal graph applications, such as natural language processing (Xu et al., 2022), molecular optimization (Fu et al., 2021), learning on point clouds (Wang et al., 2019), and improving GNN robustness against adversarial attacks (Zhang & Zitnik, 2020; Jin et al., 2020). GSL has also been employed for spatiotemporal modeling of traffic flows (Zhang et al., 2020; Tang et al., 2022a; Shang et al., 2021; Wu et al., 2019; Bai et al., 2020), irregularly sampled multivariate time series (Zhang et al., 2022), functional magnetic resonance imaging (fMRI) (El-Gazzar et al., 2021; Gazzar et al., 2022b), and sleep staging (Jia et al., 2020), but they are limited to sequences of less than 1k time steps and do not capture dynamic graph structures evolving over time.

In this study, we address the foregoing challenges by (1) leveraging S4 to enable long-range temporal modeling and (2) proposing a graph structure learning layer to learn dynamically evolving graph structures in multivariate signals. Our main contributions are:

- We propose GRAPHS4MER (Figure 1), a general end-to-end GNN architecture for spatiotemporal modeling of multivariate signals. Our model has two major advantages: (1) it leverages S4 to capture *long-range temporal dependencies* in signals and (2) it is able to *dynamically learn the underlying graph structures* in the data without a predefined graph.

- We evaluate GRAPHS4MER on three datasets with distinct data types and tasks. Our model consistently outperforms existing methods on (1) seizure detection from EEG signals, outperforming a previous GNN with self-supervised pretraining by 3.1 points in AUROC; (2) sleep staging from polysomnography signals, outpeforming existing sleep staging models

by 4.1 points in macro-F1 score; (3) traffic forecasting on PEMS-BAY benchmark, reducing MAE by 8.8% compared to existing GNNs and by 1.4% compared to Transformers.

- Qualitative interpretability analysis suggests that our GSL method learns meaningful graph structures that reflect the underlying seizure classes and sleep stages.

## 2 RELATED WORKS

**Temporal graph neural networks.** Temporal GNNs have been widely used for multivariate time series. A recent work by Gao & Ribeiro (2022) shows that existing temporal GNNs can be grouped into two categories: time-and-graph and time-then-graph. The time-and-graph framework treats a temporal graph as a sequence of graph snapshots, constructs a graph representation for each snapshot, and embeds the temporal evolution using sequence models. For example, GCRN (Seo et al., 2018) uses spectral graph convolution (Defferrard et al., 2016) for node representations and Long Short-Term Memory network (LSTM) (Hochreiter & Schmidhuber, 1997) for temporal dynamics modeling. DCRNN (Li et al., 2018b) integrates diffusion convolution with Gated Recurrent Units (GRU) (Cho et al., 2014) for traffic forecasting. Li et al. (2019) combines relational graph convolution with LSTM for path failure prediction. EvolveGCN (Pareja et al., 2020) treats graph convolutional network (GCN) parameters as recurrent states and uses an recurrent neural network (RNN) to evolve the GCN parameters. In contrast, time-then-graph framework sequentially represents the temporal evolution of node and edge attributes, and uses these temporal representations to build a static graph representation (da Xu et al., 2020; Rossi et al., 2020). Gao & Ribeiro (2022) proposes a general framework for time-then-graph, which represents the evolution of node and edge features using two RNNs independently, constructs a static graph from these representations, and encodes them using a GNN. However, most of the existing temporal GNNs are limited to short sequences and adopt RNNs or self-attention for temporal modeling, which may be suboptimal when applied to long sequences. To our knowledge, only one study has leveraged S4 for temporal graphs (Gazzar et al., 2022a). However, the learned graph is static and shared across all data points, and it is limited to fMRI data.

**Graph structure learning.** GSL aims to jointly learn a graph structure and its corresponding node (or graph) representations. Briefly, GSL methods fall into the following three categories (Zhu et al., 2021): (1) metric-based approaches that derive edge weights from node embeddings based on a metric function (Li et al., 2018a; Chen et al., 2020; Zhang & Zitnik, 2020), (2) neural-based approaches that use neural networks to learn the graph structure from node embeddings (Luo et al., 2021; Zheng et al., 2020a; Jia et al., 2020), and (3) direct approaches that treat the adjacency matrix or node embedding dictionary as learnable parameters (Jin et al., 2020; Gao et al., 2020). GSL has been applied to temporal graph data. Zhang et al. (2020) proposes learning graph adjacency matrix from traffic data through learnable parameters. Tang et al. (2022a) uses a multi-layer perceptron and a similarity measure to learn the graph structure for traffic forecasting. Shang et al. (2021) leverages a link predictor (two fully connected layers) to obtain the edge weights. Several works adaptively learn the adjacency matrix from data using learnable node embedding dictionaries (Wu et al., 2019; El-Gazzar et al., 2021; Gazzar et al., 2022a). However, these GSL approaches are limited to short sequences of less than 1k time steps and do not capture dynamically evolving graph structures.

## 3 BACKGROUND

### 3.1 STRUCTURED STATE SPACE MODEL (S4)

The Structured State Space sequence model (S4) (Gu et al., 2022) is a deep sequence model that is capable of capturing long-range dependencies. S4 has achieved state-of-the-art on pixel-level 1-D image classification, the challenging Long Range Arena benchmark for long sequence modeling, raw speech classification, and audio generation (Gu et al., 2022; Goel et al., 2022).

S4 is based on the state space model (SSM). A recent line of work (Gu et al., 2020; 2021; Gu et al., 2022) has shown that SSMs can be viewed as both convolutional neural networks (CNNs) and RNNs. The continuous-time SSM is defined as follows, which maps an 1-D signal $u(t)$ to a high dimensional latent state $x(t)$ before projecting to a 1-D output signal $y(t)$:

$$x'(t) = \mathbf{A}x(t) + \mathbf{B}u(t), \ \ y(t) = \mathbf{C}x(t) + \mathbf{D}u(t) \tag{1}$$

Equation 1 can be used as a black-box sequence representation in a deep learning model, where $\mathbf{A}$, $\mathbf{B}$, $\mathbf{C}$, and $\mathbf{D}$ are parameters learned by gradient descent. In the rest of this section, $\mathbf{D}$ is omitted for exposition because $\mathbf{D}u(t)$ can be viewed as a skip connection and can be easily computed (Gu et al., 2022).

Gu et al. (2021) shows that discrete-time SSM is equivalent to the following convolution:

$$\overline{\mathbf{K}} := (\overline{\mathbf{CB}}, \overline{\mathbf{CAB}}, ..., \overline{\mathbf{CA}}^{L-1}\overline{\mathbf{B}}), \; y = \overline{\mathbf{K}} * u \tag{2}$$

where $\overline{\mathbf{A}}$, $\overline{\mathbf{B}}$, and $\overline{\mathbf{C}}$ are the discretized matrices of $\mathbf{A}, \mathbf{B}$, and $\mathbf{C}$, respectively; $\overline{\mathbf{K}}$ is called the SSM convolution kernel; and $L$ is the sequence length.

S4 (Gu et al., 2022) is a special instantiation of SSMs that parameterizes $\mathbf{A}$ as a diagonal plus low-rank (DPLR) matrix. This parameterization has two major advantages. First, it enables fast computation of the SSM kernel $\overline{\mathbf{K}}$. Second, the parameterization includes the HiPPO matrices (Gu et al., 2020), which are a class of matrices that is capable of capturing long-range dependencies.

However, naively applying S4 for multivariate signals mixes the signal channels with a position-wise linear layer (Gu et al., 2022), which does not take into account the underlying graph structure of multivariate signals. This motivates our development of GRAPHS4MER described in Section 4.

## 3.2 GRAPH REGULARIZATION

Graph regularization encourages a learned graph to have several desirable properties, such as smoothness, sparsity, and connectivity (Chen et al., 2020; Kalofolias, 2016; Zhu et al., 2021). Let $\mathbf{X} \in \mathbb{R}^{N \times D}$ be a graph data with $N$ nodes and $D$ features. First, a common assumption in graph signal processing is that features change smoothly between adjacent nodes (Ortega et al., 2018). Given an undirected graph with adjacency matrix $\mathbf{W}$, the smoothness of the graph can be measured by the Dirichlet energy (Belkin & Niyogi, 2001):

$$\mathcal{L}_{\text{smooth}}(\mathbf{X}, \mathbf{W}) = \frac{1}{2N^2} \sum_{i,j} \mathbf{W}_{ij} ||\mathbf{X}_{i,:} - \mathbf{X}_{j,:}||^2 = \frac{1}{N^2} \text{tr}(\mathbf{X}^T \mathbf{L} \mathbf{X}) \tag{3}$$

where tr(.) denotes the trace of a matrix, $\mathbf{L} = \mathbf{D} - \mathbf{W}$ is the graph Laplacian, $\mathbf{D}$ is the degree matrix of $\mathbf{W}$. Minimizing Equation 3 therefore encourages the learned graph to be smooth. In practice, the normalized graph Laplacian $\hat{\mathbf{L}} = \mathbf{D}^{-1/2}\mathbf{L}\mathbf{D}^{-1/2}$ is used so that it is independent of node degrees.

However, simply minimizing the smoothness may result in a trivial solution $\mathbf{W} = \mathbf{0}$ (Chen et al., 2020). To avoid this trivial solution and encourage sparsity of the learned graph, additional constraints can be added (Chen et al., 2020; Kalofolias, 2016):

$$\mathcal{L}_{\text{degree}}(\mathbf{W}) = \frac{-1}{N} \mathbf{1}^T \log(\mathbf{W}\mathbf{1}), \; \mathcal{L}_{\text{sparse}}(\mathbf{W}) = \frac{1}{N^2} ||\mathbf{W}||_F^2 \tag{4}$$

where $||.||_F$ is the Frobenius norm of a matrix. Intuitively, $\mathcal{L}_{\text{degree}}$ penalizes disconnected graphs and $\mathcal{L}_{\text{sparse}}$ discourages nodes with high degrees (i.e., encourages the learned graph to be sparse).

## 4 METHODS

### 4.1 PROBLEM SETUP

In this study, we propose a general representation of multivariate signals using graphs. Let $\mathbf{X} \in \mathbb{R}^{N \times T \times M}$ be a multivariate signal, where $N$ is the number of sensors, $T$ is the sequence length, and $M$ is the input dimension of the signal (typically $M = 1$). We represent the multivariate signal as a graph $\mathcal{G} = \{\mathcal{V}, \mathcal{E}, \mathbf{W}\}$, where the set of nodes $\mathcal{V}$ corresponds to the sensors, $\mathcal{E}$ is the set of edges, and $\mathbf{W}$ is the adjacency matrix. Here, $\mathcal{E}$ and $\mathbf{W}$ are unknown and will be learned by our model.

While our formulation is general to any type of node-level or graph-level classification and regression tasks, we focus on graph-level classification (e.g., seizure detection) and node-level forecasting problems (e.g., traffic forecasting) in this work. For graph classification, the goal is to learn a function which maps an input signal to a prediction, i.e., $f : \mathbf{X} \to y \in \{1, 2, ..., C\}$, where $C$ is the number of classes. For node-level forecasting, the goal is to learn a function which predicts the next $S'$ time step signals given the previous $S$ time step signals, i.e., $f : \mathbf{X}_{:,(t-S):t,:} \to \mathbf{X}_{:,t:(t+S'),:}$.

## 4.2 Temporal modeling with S4

In this work, we leverage S4 to capture long-range temporal dependencies in signals. Naively applying S4 for multivariate signals mixes the $N$ signal channels with a position-wise linear layer (Gu et al., 2022), which may be suboptimal because it neglects the underlying graph structure of multivariate signals. Instead, we use stacked S4 layers with residual connection to embed signals in each channel (sensor) independently, resulting in an embedding $\mathbf{H} \in \mathbb{R}^{N \times T \times D}$ for each input signal $\mathbf{X}$, where $D$ is the embedding dimension.

## 4.3 Dynamic graph structure learning

**Graph structure learning layer.** To model spatial dependencies in signals (i.e., graph adjacency matrix $\mathbf{W}$), we develop a graph structure learning (GSL) layer to learn the similarities between nodes. To capture the dynamics of signals that evolve over time, our GSL layer learns a *unique* graph structure within a short time interval $r$, where $r$ is a pre-specified resolution. Instead of learning a unique graph at each time step, we choose to learn a graph over $r$ time interval because (1) aggregating information across a longer time interval can result in less noisy graphs and (2) it is more computationally efficient. For convenience, we let the predefined resolution $r$ be an integer and assume that the sequence length $T$ is divisible by $r$ without overlaps. In the remainder of this paper, we denote $n_d = \frac{T}{r}$ as the number of dynamic graphs.

For inductive settings where the task is to predict on unseen multivariate signals (e.g., seizure detection), the goal of GSL is to learn a unique graph for each multivariate signal within $r$ time interval. To obtain the edge weight between two nodes, we adopt self-attention (Vaswani et al., 2017) to allow each node to attend to all other nodes and use the attention weight as the edge weight:

$$\mathbf{Q} = \mathbf{h}^{(t)}\mathbf{M}^Q, \ \ \mathbf{K} = \mathbf{h}^{(t)}\mathbf{M}^K, \ \ \overline{\mathbf{W}}^{(t)} = \text{softmax}(\frac{\mathbf{Q}\mathbf{K}^T}{\sqrt{D}}), \text{ for } t = 1, 2, ..., n_d \quad (5)$$

Here, $\mathbf{h}^{(t)} \in \mathbb{R}^{N \times D}$ is mean-pooled S4 embeddings within time $((t-1) \times r, t \times r]$; $\mathbf{M}^Q \in \mathbb{R}^{D \times D}$ and $\mathbf{M}^K \in \mathbb{R}^{D \times D}$ are weights projecting $\mathbf{h}^{(t)}$ to query $\mathbf{Q}$ and key $\mathbf{K}$, respectively; $\overline{\mathbf{W}}^{(t)} \in \mathbb{R}^{N \times N}$ is the learned adjacency matrix within time $((t-1) \times r, t \times r]$. Equation 5 can be easily extended to multihead self-attention (Vaswani et al., 2017).

For transductive settings where all sensors form a graph (e.g., traffic forecasting), we use a learnable embedding $\mathbf{E}^{(t)} \in \mathbb{R}^{N \times d_e}$ in GSL to learn a graph for all multivariate signals within $r$ time interval (Bai et al., 2020; Wu et al., 2019):

$$\overline{\mathbf{W}}^{(t)} = \text{softmax}\big(\text{LeakyReLU}(\mathbf{E}^{(t)}\mathbf{E}^{(t)T})\big), \text{ for } t = 1, 2, ..., n_d \quad (6)$$

Here, $d_e$ is the embedding dimension, and $\mathbf{E}^{(t)}$ is initialized randomly and trained end-to-end.

There are several advantages of our GSL layer compared to prior works (Wu et al., 2019; El-Gazzar et al., 2021; Gazzar et al., 2022a). First, our GSL layer is operated on S4 embeddings rather than raw signals, which leverages the long-range dependencies learned by S4 layers. Second, in inductive settings, our GSL layer learns a dynamic graph within a short time interval $r$ for each multivariate signal rather than a universal, static graph for all multivariate signals. Third, our GSL layer is able to learn dynamically evolving graph structures over time.

To guide the graph structure learning process, we add a k-nearest neighbor (KNN) graph $\mathbf{W}_{\text{KNN}}^{(t)}$ to the learned adjacency matrix $\overline{\mathbf{W}}^{(t)}$, where each node's k-nearest neighbors are defined by cosine similarity between their respective values in $\mathbf{h}^{(t)}$, and $\mathbf{h}^{(t)}$ is the mean-pooled S4 embeddings within time $((t-1) \times r, t \times r]$:

$$\mathbf{W}^{(t)} = \epsilon \mathbf{W}_{\text{KNN}}^{(t)} + (1-\epsilon)\overline{\mathbf{W}}^{(t)} \quad (7)$$

where $\epsilon \in [0, 1)$ is a hyperparameter for the weight of the KNN graph.

To introduce graph sparsity, we prune the adjacency matrix $\mathbf{W}^{(t)}$ by removing edges whose weights are smaller than a certain threshold, i.e., $\mathbf{W}_{ij}^{(t)} = 0$ if $\mathbf{W}_{ij}^{(t)} < \kappa$, where $\kappa$ is a hyperparameter. For undirected graphs, we make the learned adjacency matrix symmetric by taking the average of the edge weights between two nodes.

**Graph regularization.** To encourage the learned graphs to have desirable graph properties, we include three regularization terms as detailed in Equations 3-4. Let $\mathbf{h}^{(t)} \in \mathbb{R}^{N \times D}$ be the mean-pooled S4 embeddings within $((t-1) \times r, t \times r]$. The regularization loss is the weighted sum of the three regularization terms and averaged across all dynamic graphs: $\mathcal{L}_{\text{reg}} = \frac{1}{n_d} \sum_{t=1}^{n_d} \alpha \mathcal{L}_{\text{smooth}}(\mathbf{h}^{(t)}, \mathbf{W}^{(t)}) + \beta \mathcal{L}_{\text{degree}}(\mathbf{W}^{(t)}) + \gamma \mathcal{L}_{\text{sparse}}(\mathbf{W}^{(t)})$, where $\alpha$, $\beta$, and $\gamma$ are hyperparameters.

### 4.4 MODEL ARCHITECTURE

The overall architecture of our model (Figure 1 and Algorithm 1 in Appendix A), GRAPHS4MER, consists of three main components: (1) stacked S4 layers with residual connection to model temporal dependencies in signals within each sensor independently, which maps raw signals $\mathbf{X} \in \mathbb{R}^{N \times T \times M}$ to S4 embedding $\mathbf{H} \in \mathbb{R}^{N \times T \times D}$, (2) a GSL layer to learn dynamically evolving adjacency matrices $\mathbf{W}^{(1)}, ..., \mathbf{W}^{(n_d)}$, and (3) GNN layers to learn spatial dependencies between sensors given the learned graph structures $\mathbf{W}^{(1)}, ..., \mathbf{W}^{(n_d)}$ and node features $\mathbf{H}$. While our model is general to any kind of GNN layers, we use an expressive GNN architecture, Graph Isomorphism Network (GIN) (Xu et al., 2019; Hu et al., 2020), in our experiments.

For graph classification tasks, a temporal pooling layer and a graph pooling layer are added to aggregate temporal and spatial representations, respectively, which is followed by a fully connected layer to produce a prediction for each multivariate signal. Whereas for node forecasting tasks, a fully connected layer is added to project the outputs of the GNN layers to the output dimension. The total loss is the sum of the regularization loss and the prediction loss. Figure 1 shows the overall model architecture, and Algorithm 1 in Appendix A shows the pseudocode for GRAPHS4MER. Source code is included as supplemental material.

## 5 EXPERIMENTS

### 5.1 EXPERIMENTAL SETUP

In this section, we briefly introduce the datasets and experimental setup in our study. For each experiment, we ran three runs with different random seeds and report mean and standard deviation of the results. See Appendix B–C for details on datasets, model training procedures, and hyperparameters.

**Seizure detection from EEGs.** We first evaluate our model on EEG-based seizure detection. We use the publicly available Temple University Hospital Seizure Detection Corpus (TUSZ) v1.5.2 (Shah et al., 2018), and follow the experimental setup of seizure detection on 60-s EEG clips as in Tang et al. (2022b). The number of EEG sensors is 19. The sampling rate of each 60-s EEG clip is 200 Hz, resulting in a sequence length of 12k time steps. The task is binary classification of detecting whether or not a 60-s EEG clip contains seizure. The resolution $r$ in GSL layer is chosen as 10-s (i.e., 2,000 time steps), which is inspired by how trained EEG readers analyze EEGs. As this is an inductive graph classification task, we adopt Equation 5 for GSL. The EEG graphs are undirected.

We compare our model performance to existing models for seizure detection, including (1) LSTM (Hochreiter & Schmidhuber, 1997), a variant of RNN with gating mechanisms; (2) Dense-CNN (Saab et al., 2020), a densely connected CNN specialized in seizure detection; (3) CNN-LSTM (Ahmedt-Aristizabal et al., 2020); and (4) Dist- and Corr-DCRNN without and with self-supervised pre-training (Tang et al., 2022b). Following prior studies, we use AUROC as the evaluation metric.

**Sleep staging from polysomnography signals.** Next, we evaluate our model on sleep staging from polysomnography (PSG) signals. PSG is used in the diagnosis of sleep disorders such as obstructive sleep apnea. We use the publicly available Dreem Open Dataset-Healthy (DOD-H) (Guillot et al., 2020). We randomly split the data by 60/20/20 into train/validation/test sets, where each split consists of data from distinct subjects. The number of PSG sensors is 16. The sampling rate is 250 Hz, and thus each 30-s signal has 7,500 time steps. The task is to classify each 30-s PSG signal as one of the five sleep stages: wake, rapid eye movement (REM), non-REM sleep stages N1, N2, and N3. The resolution $r$ is 10-s (i.e., 2.5k time steps). This is also an inductive graph classification task, and thus we adopt Equation 5 for GSL. The PSG graphs are undirected.

Baseline models include existing sleep staging models that achieved state-of-the-art on DOD-H dataset, SimpleSleepNet (Guillot et al., 2020) and RobustSleepNet (Guillot & Thorey, 2021), both

of which are based on GRUs (Cho et al., 2014) and attention mechanisms. We also include the traditional sequence model LSTM (Hochreiter & Schmidhuber, 1997) as a baseline. For fair comparisons between the baselines and GRAPHS4MER, we trained SimpleSleepNet and RobustSleepNet using their open sourced code[1] and setting the temporal context to be one 30-s PSG signal. Similar to Guillot & Thorey (2021), we use macro-F1 score and Cohen's Kappa as the evaluation metrics.

**Traffic forecasting.** Finally, we evaluate our model on forecasting of traffic flows on the public PEMS-BAY dataset (Li et al., 2018b). The traffic dataset is collected by California Transportation Agencies (CalTrans) Performance Measurement System (PeMS) from January 1st 2017 to May 31st 2017. We follow the same experimental setup as that in Li et al. (2018b). The number of traffic sensors is 325. The traffic speed readings are aggregated into 5-min windows and the data are split by 70/10/20 into train/validation/test sets. The task is to predict the future 1 hour traffic flow (i.e., prediction horizon = 12 time steps) given the previous 1 hour traffic flow (12 time steps). The resolution $r$ is set to be 12 time steps. This is a transductive node-level forecasting task, therefore we adopt Equation 6 for graph structure learning. The traffic graph is directed.

We compare our model performance to existing traffic forecasting models: (1) DCRNN (Li et al., 2018b), (2) Graph WaveNet (Wu et al., 2019), (3) Graph Multi-Attention Network (GMAN) (Zheng et al., 2020b), (4) Traffic Transformer (Cai et al., 2020), (5) Spacetimeformer (Grigsby et al., 2022), and (6) Adaptive Graph Spatial-Temporal Transformer Network (ASTTN) (Feng & Tassiulas, 2022). DCRNN, Graph WaveNet, and GMAN are temporal GNNs, whereas the rest are Transformer-based models. Following previous works, we use mean absolute error (MAE), root mean squared error (RMSE), and mean absolute percentage error (MAPE) as the evaluation metrics.

## 5.2 EXPERIMENTAL RESULTS

**Seizure detection from EEG signals.** Table 1 shows our model performance on EEG-based seizure detection and its comparison to existing models. GRAPHS4MER outperforms the previous state-of-the-art, Dist-DCRNN with pretraining, by 3.1 points in AUROC. Notably, Dist-DCRNN was pretrained using a self-supervised task (Tang et al., 2022b), whereas GRAPHS4MER was trained from scratch.

Table 1: **Results of EEG-based seizure detection on TUSZ dataset.** Baseline model results are cited from Tang et al. (2022b). Best and second best results are **bolded** and underlined, respectively.

| Model | AUROC |
|---|---|
| LSTM (Hochreiter & Schmidhuber, 1997) | $0.715 \pm 0.016$ |
| Dense-CNN (Saab et al., 2020) | $0.796 \pm 0.014$ |
| CNN-LSTM (Ahmedt-Aristizabal et al., 2020) | $0.682 \pm 0.003$ |
| Dist-DCRNN w/o Pretraining (Tang et al., 2022b) | $0.793 \pm 0.022$ |
| Corr-DCRNN w/o Pretraining (Tang et al., 2022b) | $0.804 \pm 0.015$ |
| Dist-DCRNN w/ Pretraining (Tang et al., 2022b) | $\underline{0.875 \pm 0.016}$ |
| Corr-DCRNN w/ Pretraining (Tang et al., 2022b) | $0.850 \pm 0.014$ |
| GRAPHS4MER (ours) | $\mathbf{0.906 \pm 0.012}$ |

**Sleep staging from PSG signals.** Table 2 compares our model performance to existing sleep staging models on DOD-H dataset. GRAPHS4MER outperforms RobustSleepNet, a specialized sleep staging model, by 4.1 points in macro-F1. It is important to note that both SimpleSleepNet and RobustSleepNet preprocess the PSG signals using short-time Fourier transform, whereas our model directly operates on raw PSG signals without the need of preprocessing.

**Traffic forecasting.** Table 3 shows the performance of our model and existing traffic forecasting models on PEMS-BAY. We observe that GRAPHS4MER outperforms temporal GNNs (DCRNN, Graph WaveNet, GMAN) by a large margin and performs better than Transformer-based models (ASTTN and Spacetimeformer) in MAE.

Notably, the three tasks cover a wide range of sequence lengths ranging from 12 to 12k time steps. GRAPHS4MER outperforms previous state-of-the-art on all of the three tasks, suggesting that it is capable of capturing long-range temporal dependencies across a wide range of sequence lengths.

---

[1]SimpleSleepNet: `https://github.com/Dreem-Organization/dreem-learning-open`;
RobustSleepNet: `https://github.com/Dreem-Organization/RobustSleepNet`

Table 2: **Results of sleep staging on DOD-H dataset.** Best and second best results are **bolded** and underlined, respectively.

| Model | Macro-F1 | Kappa |
|---|---|---|
| LSTM (Hochreiter & Schmidhuber, 1997) | $0.609 \pm 0.034$ | $0.539 \pm 0.046$ |
| SimpleSleepNet (Guillot et al., 2020) | $0.720 \pm 0.001$ | $0.703 \pm 0.013$ |
| RobustSleepNet (Guillot & Thorey, 2021) | $\underline{0.777 \pm 0.007}$ | $\underline{0.758 \pm 0.008}$ |
| GRAPHS4MER (ours) | $\mathbf{0.818 \pm 0.008}$ | $\mathbf{0.802 \pm 0.014}$ |

Table 3: **Results of traffic forecasting on PEMS-BAY dataset.** Baseline model results are cited from the literature. Best and second best results are **bolded** and underlined, respectively.

| Model | MAE | RMSE | MAPE |
|---|---|---|---|
| DCRNN (Li et al., 2018b) | 2.07 | 4.74 | 4.90% |
| Graph WaveNet (Wu et al., 2019) | 1.95 | 4.52 | 4.63% |
| GMAN (Zheng et al., 2020b) | 1.86 | 4.32 | 4.31% |
| Traffic Transformer (Cai et al., 2020) | 1.77 | 4.36 | 4.29% |
| Spacetimeformer (Grigsby et al., 2022) | 1.73 | **3.79** | **3.85%** |
| ASTTN (Feng & Tassiulas, 2022) | $\underline{1.72}$ | 4.02 | 3.98% |
| GRAPHS4MER (ours) | $\mathbf{1.696 \pm 0.008}$ | $\underline{3.989 \pm 0.023}$ | $3.86\% \pm 0.02\%$ |

**Effect of GSL.** As GSL is an important component of GRAPHS4MER, we investigate the effect of GSL on the model performance. Table 4 shows the model performance when we remove the GSL layer in GRAPHS4MER and use predefined graphs. For seizure detection on TUSZ dataset, we use the predefined distance-based EEG graph as in Tang et al. (2022b); for traffic forecasting on PEMS-BAY, we use the predefined traffic sensor graph as in Li et al. (2018b). DOD-H dataset has no predefined graph available, and thus we use a KNN graph obtained based on cosine similarity between nodes using S4 embeddings (K = 3; same as GRAPHS4MER). We observe that GRAPHS4MER using GSL outperforms GRAPHS4MER with predefined graph on the TUSZ and DOD-H datasets, but not on the PEMS-BAY dataset. This suggests that sensor locations used for constructing the predefined traffic sensor graph (Li et al., 2018b) are critical for traffic forecasting. Nevertheless, a major advantage of our GSL method is that it does not require prior knowledge about the graph structure, which would be particularly useful when sensor locations are not available.

Table 4: Comparison of predefined graphs vs GSL. Best results are **bolded**.

| Model | TUSZ (AUROC) | DOD-H (Macro-F1) | PEMS-BAY (MAE) |
|---|---|---|---|
| GRAPHS4MER w/ predefined graph (w/o GSL) | $0.899 \pm 0.010$ | $0.765 \pm 0.016$ | $\mathbf{1.629 \pm 0.003}$ |
| GRAPHS4MER (w/ GSL) | $\mathbf{0.906 \pm 0.012}$ | $\mathbf{0.818 \pm 0.008}$ | $1.696 \pm 0.008$ |

**Ablations.** We perform ablation studies to investigate the importance of (1) graph-based representation, where we remove the GSL and GNN layers in our model and (2) S4 encoder, where we replace the S4 layers with GRUs (Cho et al., 2014). Table 5 shows the ablation results. There are several important observations. First, GRAPHS4MER consistently outperforms S4, suggesting the effectiveness of representing multivariate signals as graphs. Second, when S4 layers are replaced with GRUs, the model performance drops by a large margin, particularly on TUH and DOD-H datasets with long sequences (12k and 7.5k time steps, respectively). This indicates the effectiveness of S4 in modeling long-term temporal dependencies.

**Effect of temporal resolution.** We examine the effect of temporal resolution $r$ in Figure 3 in Appendix D. We observe that dynamically varying graph structures are more useful for inductive tasks (i.e., seizure detection and sleep staging) than the transductive traffic forecasting task.

**Visualization of graphs.** To investigate if the learned graphs are interpretable, we visualize mean adjacency matrices for EEG and PSG signals in the correctly predicted test samples in Figure 2, grouped by seizure classes and sleep stages. For EEG, we observe that the differences between generalized seizure and non-seizure adjacency matrices (Figure 2e) are larger than the differences between focal seizure and non-seizure adjacency matrices (Figure 2d). This suggests that generalized seizures disrupt normal brain connectivity more than focal seizures, which is consistent with

Table 5: **Ablation results.** For S4, we remove the GSL and GNN layers. For GRAPHS4MER w/o S4, we replace S4 with GRUs. Best and second best results are **bolded** and underlined, respectively.

| Model | TUSZ (AUROC) | DOD-H (Macro-F1) | PEMS-BAY (MAE) |
|---|---|---|---|
| S4 | $\underline{0.824 \pm 0.011}$ | $\underline{0.778 \pm 0.009}$ | $2.262 \pm 0.004$ |
| GRAPHS4MER w/o S4 | $0.705 \pm 0.095$ | $0.634 \pm 0.061$ | $\underline{1.723 \pm 0.006}$ |
| GRAPHS4MER | $\mathbf{0.906 \pm 0.012}$ | $\mathbf{0.818 \pm 0.008}$ | $\mathbf{1.696 \pm 0.008}$ |

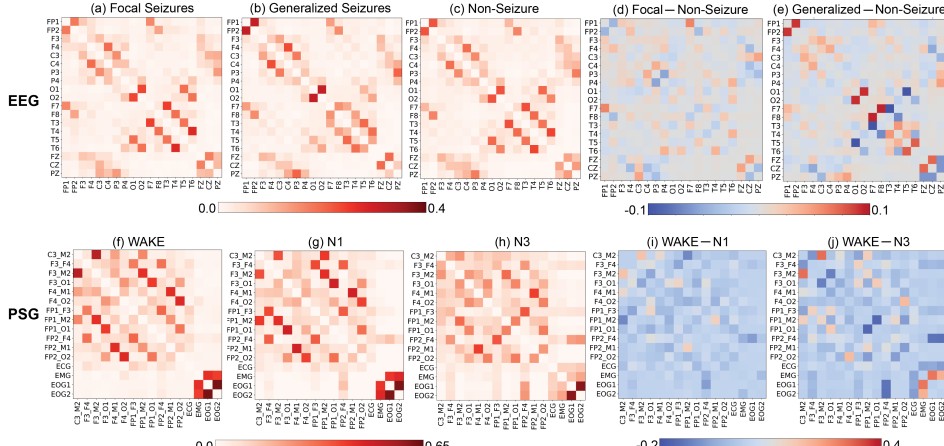

Figure 2: **Top**: Mean adjacency matrix for EEG in correctly predicted test samples for **(a)** focal seizures, **(b)** generalized seizures, and **(c)** non-seizure EEG clips. **(d)** Difference between focal seizure and non-seizure adjacency matrices. **(e)** Difference between generalized seizure and non-seizure adjacency matrices. **Bottom**: Mean adjacency matrix for PSG in correctly predicted test samples for sleep stage **(f)** WAKE, **(g)** N1, and **(h)** N3. **(i)** Difference between WAKE and N1 adjacency matrices. **(j)** Difference between WAKE and N3 adjacency matrices. Self-edges (i.e., diagonal) are not shown here.

the literature that brain functional connectivity changes significantly in generalized seizures (Sun et al., 2021). For PSG, we observe that the differences between N3 and wake (Figure 2j) are larger than the differences between N1 and wake (Figure 2i). This pattern is expected given that N1 is the earliest sleep stage, whereas N3 is the deep sleep stage and is associated with slow brain waves that are not present in other sleep stages (Academy Sleep Medicine, 2007). Mean adjacency matrices for sleep stages N2 and REM are shown in Appendix E and Figure 4. In addition, we visualize the learned adjacency matrix for traffic forecasting in Appendix F and Figure 5. We also show example dynamic graphs for an EEG clip in Appendix G and Figure 6.

## 6 CONCLUSION AND FUTURE WORK

In conclusion, we propose a novel graph structure learning method to learn dynamically evolving graph structures in multivariate signals, and present GRAPHS4MER, a general GNN architecture integrating Structured State Spaces model and graph structure learning for spatiotemporal modeling of multivariate signals. Our method sets new state-of-the-art on seizure detection, sleep staging, and traffic forecasting, and learns meaningful graph structures that reflect seizure classes and sleep stages. Importantly, our method does not require prior knowledge about the graph structure, and thus would be particularly useful for applications where graph structures cannot be easily predefined. Our study opens exciting future directions for graph-based spatiotemporal modeling of multivariate signals, including (1) leveraging domain knowledge for improved graph structure learning, (2) investigating other ways of combining S4 and GSL to further improve long-range graph structure learning, and (3) applying to other types of time series.

ETHICS STATEMENT

The Temple University Hospital EEG Seizure Corpus (TUSZ) is publicly available[2] and de-identified with full IRB approval (Shah et al., 2018). The DOD-H dataset is publicly available[3], deidentified, and was approved by the French Committees of Protection of Persons (Guillot et al., 2020). The PEMS-BAY dataset is publicaly available[4] and no human or animal subject is involved (Li et al., 2018b). No conflict of interest is reported from the authors. No harmful insights are provided by the models. While we show that our methods could provide improved performance for seizure detection and sleep staging, additional model validations are needed before they can be used in real-world clinical settings, such as (a) external validation on datasets from multiple institutions and (b) validation on subjects with different demographics.

REPRODUCIBILITY STATEMENT

All the datasets used in our study are publicly available. Details about datasets, model training procedures, and hyperparameters are provided in the Appendix. Source code is included in supplemental materials.

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

APPENDIX

# A PSEUDOCODE FOR GRAPHS4MER

Algorithm 1 shows the psuedocode for GraphS4mer. Note that $n_d = \frac{T}{r}$ is the number of dynamic graphs, where $T$ is the sequence length and $r$ is the pre-specified resolution. While there is a for loop over $t = 1, ..., n_d$, the code implementation can be done in batches to eliminate the need of a for loop.

---

**Algorithm 1:** Pseudocode for GraphS4mer.

---

**Input**      : multivariate signal $\mathbf{X}$, label $\mathbf{y}$
**Parameter:** model weights $\theta$, including weights for S4, GSL, GNN, and linear layers
**Output**     : prediction $\hat{\mathbf{y}}$, learned graph adjacency matrices $\mathbf{W}^{(1)}, ..., \mathbf{W}^{(n_d)}$

1 Randomly initialize model weights $\theta$
2 **while** *not converged* **do**
3    $\mathcal{L}_{\text{reg}} \leftarrow 0$
4    `// step 1.  get S4 embeddings`
5    $\mathbf{H} \leftarrow \text{S4}(\mathbf{X})$
6    **for** $t = 1, 2, ..., n_d$ **do**
7      $\mathbf{h}^{(t)} \leftarrow \text{mean-pool}(\mathbf{H}_{:,((t-1)\times r):(t\times r),:})$ `// mean-pool along temporal dim`
8      `// step 2.  graph structure learning (GSL)`
9      Construct KNN graph, $\mathbf{W}^{(t)}_{\text{KNN}}$, based on cosine similarity between nodes using $\mathbf{h}^{(t)}$
10      **if** *inductive setting* **then**
11        $\mathbf{W}^{(t)} \leftarrow \text{GSL}(\mathbf{h}^{(t)}, \mathbf{W}^{(t)}_{\text{KNN}})$ using Eqn. 5 and Eqn. 7 `// GSL in inductive setting`
12      **else**
13        $\mathbf{W}^{(t)} \leftarrow \text{GSL}(\mathbf{W}^{(t)}_{\text{KNN}})$ using Eqn. 6 and Eqn. 7 `// GSL in transductive setting`
14      **end if**
15      $\mathcal{L}_{\text{reg}} \leftarrow \mathcal{L}_{\text{reg}} + \alpha \mathcal{L}_{\text{smooth}}(\mathbf{h}^{(t)}, \mathbf{W}^{(t)}) + \beta \mathcal{L}_{\text{degree}}(\mathbf{W}^{(t)}) + \gamma \mathcal{L}_{\text{sparse}}(\mathbf{W}^{(t)}))$ using Eqn. 3–4 `// graph regularization loss`
16      `// step 3.  get node embeddings from GNN`
17      **if** *graph classification task* **then**
18        $\mathbf{z}^{(t)} \leftarrow \text{GNN}(\mathbf{h}^{(t)}, \mathbf{W}^{(t)})$ `// $z^{(t)}$ has shape $(N, 1, D_{hidden})$`
19      **else**
20        $\mathbf{z}^{(t)} \leftarrow \text{GNN}(\mathbf{H}_{:,((t-1)\times r):(t\times r),:}, \mathbf{W}^{(t)})$ `// $z^{(t)}$ has shape $(N, r, D_{hidden})$`
21      **end if**
22    **end for**
23    $\mathbf{Z} \leftarrow \text{concat}(\mathbf{z}^{(1)}, \mathbf{z}^{(2)}, ..., \mathbf{z}^{(n_d)})$ `// concatenate along temporal dim`
24    `// step 4.  pooling and fully connected layers`
25    **if** *graph classification task* **then**
26      $\mathbf{Z} \leftarrow \text{graph-pool}(\text{temporal-pool}(\mathbf{Z}))$
27      $\hat{\mathbf{y}} \leftarrow \text{Linear}(\mathbf{Z})$
28      $\mathcal{L}_{\text{pred}} \leftarrow \mathcal{L}_{\text{cross-entropy}}(\hat{\mathbf{y}}, \mathbf{y})$ `// loss for classification`
29    **else**
30      $\hat{\mathbf{y}} \leftarrow \text{Linear}(\mathbf{Z})$
31      $\mathcal{L}_{\text{pred}} \leftarrow \mathcal{L}_{\text{MAE}}(\hat{\mathbf{y}}, \mathbf{y})$ `// loss for forecasting`
32    **end if**
33    $\mathcal{L} \leftarrow \mathcal{L}_{\text{pred}} + \frac{1}{n_d}\mathcal{L}_{\text{reg}}$ `// total loss`
34    Back-propagate $\mathcal{L}$ to update model weights $\theta$
35 **end while**

---

## B  DETAILS OF DATASETS

**Temple University Hospital Seizure Detection Corpus (TUSZ).** We use the publicly available Temporal University Hospital Seizure Detection Corpus (TUSZ) v1.5.2. for seizure detection (Shah et al., 2018). We follow the same experimental setup as in Tang et al. (2022b). The TUSZ train set is divided into train and validation splits with distinct patients, and the TUSZ test set is held-out for model evaluation. The following 19 EEG channels are included: FP1, FP2, F3, F4, C3, C4, P3, P4, O1, O2, F7, F8, T3, T4, T5, T6, FZ, CZ, and PZ. Because the EEG signals are sampled at different sampling rate, we resample all the EEG signals to 200 Hz. Following a prior study (Tang et al., 2022b), we also exclude 5 patients in the TUSZ test set who appear in both the TUSZ train and test sets. The EEG signals are divided into 60-s EEG clips without overlaps, and the task is to predict whether or not an EEG clip contains seizure. Table 6 shows the number of EEG clips and patients in the train, validation, and test splits.

Table 6: Number of EEG clips and patients in train, validation, and test splits of TUSZ dataset.

| Train Set | | Validation Set | | Test Set | |
|---|---|---|---|---|---|
| EEG Clips (% Seizure) | Patients (% Seizure) | EEG Clips (% Seizure) | Patients (% Seizure) | EEG Clips (% Seizure) | Patients (% Seizure) |
| 38,613 (9.3%) | 530 (34.0%) | 5,503 (11.4%) | 61 (36.1%) | 8,848 (14.7%) | 45 (77.8%) |

**Dreem Open Dataset-Healthy (DOD-H).** We use the publicly available Dreem Open Dataset-Healthy (DOD-H) for sleep staging (Guillot et al., 2020). The DOD-H dataset consists of overnight PSG sleep recordings from 25 volunteers. The PSG signals are measured from 12 EEG channels, 1 electromyographic (EMG) channel, 2 electrooculography (EOG) channels, and 1 electrocardiogram channel using a Siesta PSG device (Compumedics). All the signals are sampled at 250 Hz. Following the standard AASM Scoring Manual and Recommendations (Rosen et al., 2018), each 30-s PSG signal is annotated by a consensus of 5 experienced sleep technologists as one of the five sleep stages: wake (W), rapid eye movement (REM), non-REM sleep stages N1, N2, and N3. We randomly split the PSG signals by 60/20/20 into train/validation/test splits, where each split has distinct subjects. Table 7 shows the number of 30-s PSG clips, the number of subjects, and the five sleep stage distributions.

Table 7: Number of subjects and 30-s PSG clips in the train, validation, and test splits of DOD-H dataset.

| | Subjects | 30-s PSG Clips | | | | | |
|---|---|---|---|---|---|---|---|
| | | Total | Wake | N1 | N2 | N3 | REM |
| Train Set | 15 | 14,823 | 1,839 | 925 | 6,965 | 2,015 | 3,079 |
| Validation Set | 5 | 5,114 | 480 | 254 | 2,480 | 990 | 910 |
| Test Set | 5 | 4,725 | 718 | 326 | 2,434 | 509 | 738 |

**PEMS-BAY traffic dataset.** The PEMS-BAY dataset is collected by California Transportation Agencies (CalTrans) Performance Measurement System (PeMS). We follow the same experimental setup as in Li et al. (2018b). Specifically, 325 sensors in the Bay Area are selected with 6 months of data ranging from January 1st to May 31st, 2017. The traffic speed readings are aggregated into 5-min windows. The total number of observed traffic data points is 16,937,179. 70% of the data are used for training, 20% are used for testing, and the remaining 10% are used for validation.

## C  DETAILS OF MODEL TRAINING PROCEDURES AND HYPERPARAMETERS

Model training was accomplished using the AdamW optimizer (Loshchilov & Hutter, 2019) in PyTorch on a single NVIDIA A100 GPU. All experiments were run for three runs with different random seeds. Cosine learning rate scheduler with 5-epoch warm start was used (Loshchilov & Hutter, 2017). Model training was early stopped when the validation loss did not decrease for 20 consecutive epochs. We performed the following hyperparameter search on the validation set: (1) initial learning rate within range [1e-4, 1e-2]; (2) dropout rate in S4 and GNN layers within range [0.1, 0.5]; (3) hidden dimension of S4 and GNN layers within range {64, 128, 256}; (4) number of S4

layers within range $\{2, 3, 4\}$; (5) S4 bidirectionality; (6) number of GNN layers within range $\{1, 2\}$; (7) graph pooling within range {mean-pool, max-pool, sum-pool} (only for graph classification tasks); (8) value of $\kappa$ threshold for graph pruning by keeping the top $\kappa\%$ edges (within range $\{1, 2, 3, 4, 5, 6, 7, 8, 9, 10\}$) or keeping edges whose weights $> \kappa$ (within range $\{0.1, 0.2, 0.3, 0.4, 0.5\}$); (9) KNN graph with $K \in \{2, 3\}$; (10) weight of KNN graph $\epsilon \in [0.3, 0.6]$; (11) $\alpha, \beta$, and $\gamma$ weights in graph regularization within range $[0, 1]$; (12) learnable embedding ($\mathbf{E}^{(t)}$ in Equation 6) dimension within range $\{10, 12, 16\}$ (only for traffic forecasting).

**Model training and hyperparameters for seizure detection on TUSZ dataset.** As there are many more negative samples in the dataset, we undersampled the negative examples in the train set during training. We used binary cross-entropy loss as the loss function. The models were trained for 100 epochs with an initial learning rate of 8e-4. The batch size was 4; dropout rate was 0.1; hidden dimension was 128; number of stacked S4 layers was 4; S4 layers were unidirectional; number of GNN layers was 1; graph pooling was max-pool and temporal pooling was mean-pool; graph pruning was done by setting a threshold of 0.1, where edges whose edge weights $<= 0.1$ were removed; $K = 2$ for KNN graph and the KNN graph weight $\epsilon$ was 0.6; $\alpha, \beta$, and $\gamma$ weights were all set to 0.05. This results in 265k trainable parameters in GRAPHS4MER. Best model was picked based on the highest AUROC on the validation set.

**Model training and hyperparameters for sleep staging on DOD-H dataset.** As the DOD-H dataset is highly imbalanced, we undersampled the majority classes in the train set during training. We used cross-entropy loss as the loss function. The models were trained for 100 epochs with an initial learning rate of 1e-3. The batch size was 4; dropout rate was 0.4; hidden dimension was 128; number of stacked S4 layers was 4 and S4 layers were unidirectional; number of GNN layers was 1; graph pooling was sum-pool and temporal pooling was mean-pool; graph pruning was done by setting a threshold of 0.1; $K = 3$ for KNN graph and the weight for KNN graph $\epsilon$ was 0.6; $\alpha, \beta$, and $\gamma$ weights were all set to 0.2. This results in 266k trainable parameters in GRAPHS4MER. Best model was picked based on the highest macro-F1 score on the validation set.

**Model training and hyperparameters for traffic forecasting on PEMS-BAY.** Following Li et al. (2018b), we used mean absolute error (MAE) as the loss function. The models were trained for 100 epochs with an initial learning rate of 1e-3. The batch size was 32; dropout rate was 0.1; hidden dimension was 256; number of stacked S4 layers was 4 and S4 layers were bidirectional; number of GNN layers was 2; graph pruning was done by keeping the top 2% edges in the graph; $K = 3$ for KNN graph and the weight for KNN graph $\epsilon$ was 0.5; $\alpha, \beta$, and $\gamma$ weights were 0.1, 0.2, and 0.2, respectively; the learnable embedding dimension was 16. This results in 865k trainable parameters in GRAPHS4MER. Best model was picked based on the lowest MAE on the validation set.

## D    EFFECT OF TEMPORAL RESOLUTION

To examine the effect of temporal resolution $r$ on model performance, we show the performance versus different values of $r$ in Figure 3. We observe that for inductive tasks (i.e., seizure detection and sleep staging), smaller value of $r$ tends to result in higher performance, which suggests that capturing dynamically varying graph structures is useful for inductive tasks. In contrast, for transductive traffic forecasting task, larger value of $r$ tends to give better performance, which suggests that learning a static graph is sufficient for the traffic forecasting task.

## E    VISUALIZATION OF PSG GRAPHS FOR FIVE SLEEP STAGES

Figure 4 shows mean adjacency matrices for PSG signals in correctly predicted test samples grouped by five sleep stages. We observe that N3 differs from wake stage more than N1 (also see Figure 2). Moreover, in REM stage, the EMG channel has very weak connection to all other channels, which is expected given that one experiences muscle paralysis in REM stage.

## F    VISUALIZATION OF TRAFFIC SENSOR GRAPH

Figure 5 shows the mean adjacency matrix for traffic forecasting on PEMS-BAY dataset inferred by our GSL layer, as well as the adjacency matrix in Li et al. (2018b) predefined by sensor locations.

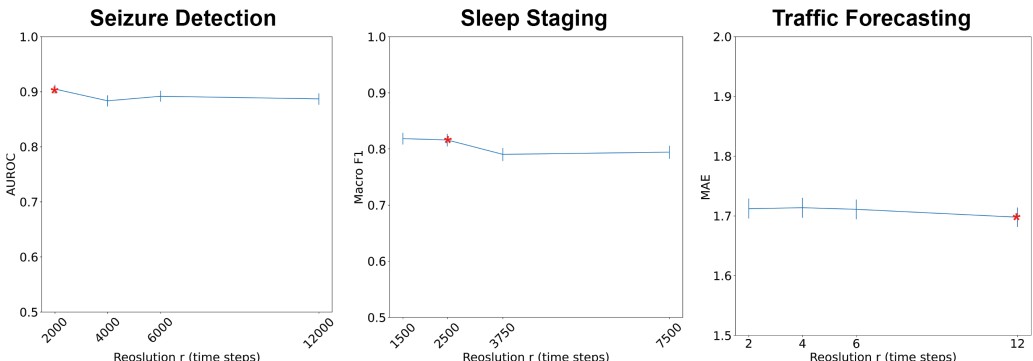

Figure 3: Model performance versus temporal resolution using the run with median performance. For convenience, we assume that temporal resolution is chosen so that the sequence length is divisible by the resolution. Asterisk indicates the temporal resolution used to report results for GRAPHS4MER in Table 1–5. Error bars indicate 95% confidence intervals obtained using bootstrapping with 5,000 replicates with replacement.

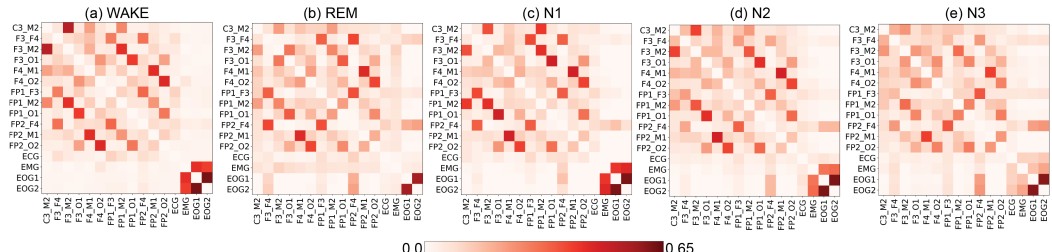

Figure 4: Mean adjacency matrix for PSG signals for five sleep stages in correctly predicted test samples. Self-edges (i.e., diagonal) are not shown here.

The learned adjacency matrix (Figure 5a) does not exhibit block structures as in the predefined adjacency matrix (Figure 5b), which could be the reason why learning the traffic sensor graph results in lower performance than the predefined sensor graph (Table 4). This also suggests that future work should explore integrating prior knowledge of the graph in the GSL layer.

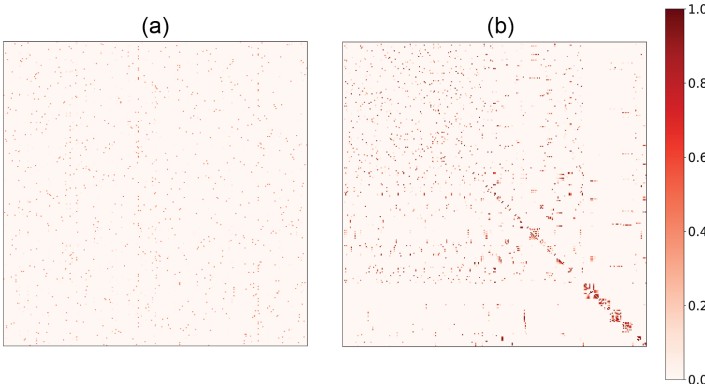

Figure 5: **(a)** Mean adjacency matrix for traffic forecasting inferred by our GSL layer. **(b)** Adjacency matrix in Li et al. (2018b) constructed based on traffic sensor locations. Self-edges (i.e., diagonal) are not shown here.

## G  VISUALIZATION OF DYNAMIC EEG GRAPHS

In Figure 6, we visualize dynamic EEG graphs in an example EEG clip in the TUSZ test set, inferred by the GSL layer. In the beginning of the EEG clip (Figures 6a–6d), the seizure starts as a focal seizure and the corresponding EEG graphs are sparse. Starting at the last 2-s of Figure 6e, the seizure starts to spread. In the last 10-s of this EEG clip (Figure 6f), the seizure spreads to all areas of the brain and the EEG graph becomes denser with more edges than the earlier EEG graphs. This change in graph connectivity reflects the dynamics of the seizure.

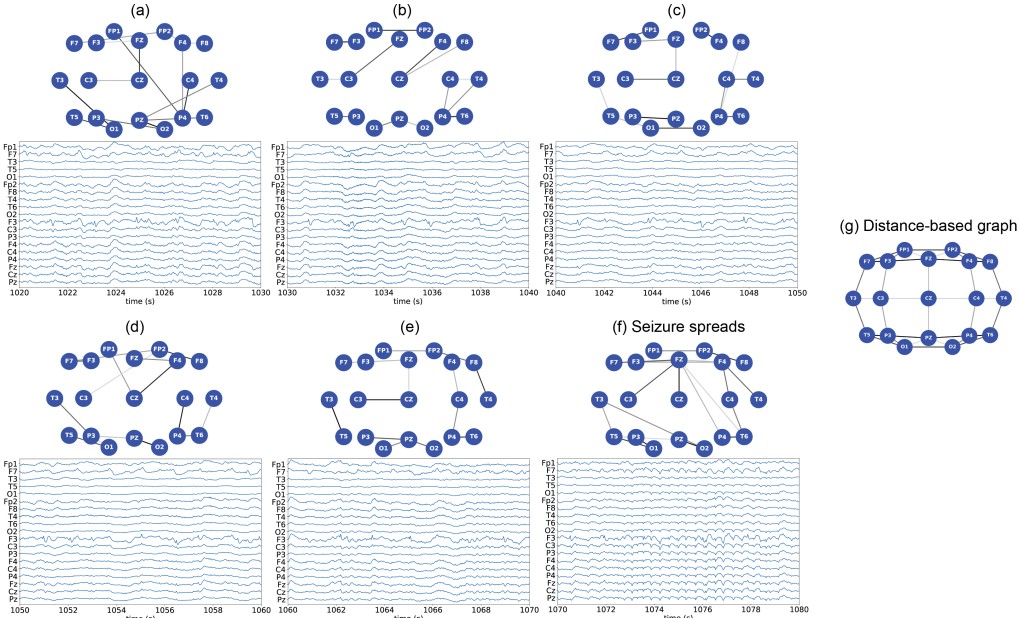

Figure 6: **Example EEG graphs of a 60-s EEG clip that has seizure in the test set**. In each subfigure in **(a)–(f)**, top panel shows the inferred graph from the GSL layer and bottom panel shows the raw EEG signals. **(g)** Distance-based graph in Tang et al. (2022b), defined based on physical distances between EEG sensors in the standard 10-20 EEG placement system.

