# OpenReview forum: "Spatiotemporal Modeling of Multivariate Signals with Graph Neural Networks and Structured State Space Models"
_ICLR.cc/2023/Conference — Submitted to ICLR 2023_

### Official Review · Reviewer_FDcr · 2022-10-24

**Confidence:** 4
**Correctness:** 4
**Technical Novelty And Significance:** 2
**Empirical Novelty And Significance:** 2
**Recommendation:** 5

**Clarity, Quality, Novelty And Reproducibility:**

Clarity: Good

Quality: Fair

Novelty: Poor

Reproducibility: Fair


**Strength And Weaknesses:**

Strength

*The authors have carefully designed a model for mining space-time relations.

*The authors evaluated the model performance on several datasets.

Weaknesses

*The innovation of the paper is insufficient. The proposed model is just a simple combination of some existing technologies. For example, the S4 and the attention mechanism in transformer are both existing technologies.

*The idea of graph structure learning exists widely in many literatures. What is the difference between the proposed method and the existing graph structure learning[1-2]?

*Explainability oriented visualization is not comprehensive. For example, there seems to be no visualization of sleep stages and traffic data. In addition, there are many edges in Figure 2 (a). But this stage does not seem to be the spread stage of epilepsy.

[1] A unified structure learning framework for graph attention networks. Neurocomputing.2022
[2] GraphSleepNet: Adaptive Spatial-Temporal Graph Convolutional Networks for Sleep Stage Classification. IJCAI.2020




**Summary Of The Paper:**

The authors present a GNN architecture combining Structured State Spaces model and graph structure learning for spatiotemporal modeling of multivariate signals.

The proposed model has two major advantages: (1) it leverages S4 to capture long-range temporal dependencies in signals and (2) it is able to dynamically learn the underlying graph structures in the data without a predefined graph.

**Summary Of The Review:**

The performance of the model is evaluated on several datasets. The model performs better than multiple baseline methods. However, the noveltyof the manuscript is very limited, which causes my concern.

---

> ### Author Response · Authors · 2022-11-17
> **Responses to Reviewer FDcr (Part 1)**
>
> We thank the reviewer for the detailed comments. Responses to specific comments are listed below.
>
> 1. *"The innovation of the paper is insufficient. The proposed model is just a simple combination of some existing technologies. For example, the S4 and the attention mechanism in transformer are both existing technologies."*
>
> We do not believe that our work is a simple combination of existing techniques. Instead, we have made novel contributions by (1) proposing a graph structure learning (GSL) method that learns dynamically evolving graph structures of multivariate signals, (2) developing a general GNN architecture leveraging Structured State Space model and GSL for graph-based spatiotemporal modeling of multivariate signals, (3) empirically showing that our proposed method consistently outperforms previous SOTA on three diverse data types and tasks, and that our GSL method learns meaningful graph structures. All of these contributions are new to the GNN and time-series community, as supported by Reviewer aQfe that “capturing and utilizing dynamic graph structures over time is an important yet largely overlooked topic in graph-based representation learning” and by Reviewer dUxW that “originality is in the proposed methodological insights, and the fact that the global pipeline is tested in three different situations which are relevant, difficult and interesting”.
>
> 2. *"The idea of graph structure learning exists widely in many literatures. What is the difference between the proposed method and the existing graph structure learning[1-2]? [1] A unified structure learning framework for graph attention networks. Neurocomputing.2022 [2] GraphSleepNet: Adaptive Spatial-Temporal Graph Convolutional Networks for Sleep Stage Classification. IJCAI.2020"*
>
> The differences between graph structure learning (GSL) methods in [1-2] and the GSL method in our work are as follows.
>
> [1] proposed to use metric learning (e.g., cosine similarity) and positive pointwise mutual information (PPMI) based on co-occurrence of two nodes in the data for GSL. The key differences between [1] and our work are: (a) the GSL method in [1] is designed for non-temporal data, whereas ours is designed for temporal data and is able to learn dynamic graph structures evolving over time; (b) the graph structure learned in [1] is shared across all data points, whereas ours dynamically learns a graph within a temporal interval for each multivariate signal (Equations 5-6). Therefore, our proposed GSL method is more suitable for our intended use cases for modeling multivariate signals, and has the advantage of capturing dynamically evolving graph structures in multivariate signals.
>
> [2] utilized a learnable weight vector of the same dimension as the node features to learn the graph structure for sleep staging. The key differences are (a) only a single graph is learned for each multivariate signal in [2], whereas our GSL method (Equation 5) learns dynamically evolving graphs for each multivariate signal; (b) graph regularization loss in [2] includes $L_{\text{smooth}}$ and $L_{\text{sparse}}$ (Equations 3 and second part of Equation 4), whereas our graph regularization loss includes an additional term $L_{\text{degree}}$ (first part of Equation 4) to discourage disconnected graphs; (c) GSL in [2] is only applied to sleep staging, whereas we evaluate our methods on three diverse tasks and show superior performance over previous state-of-the-art models.

---

> ### Author Response · Authors · 2022-11-17
> **Responses to Reviewer FDcr (Part 2)**
>
> 3. *"Explainability oriented visualization is not comprehensive. For example, there seems to be no visualization of sleep stages and traffic data. In addition, there are many edges in Figure 2 (a). But this stage does not seem to be the spread stage of epilepsy."*
>
> We have now included comprehensive graph visualization analyses as follows. In the updated Figure 2, we visualize the mean adjacency matrices for EEG signals grouped by seizure classes (focal seizure, generalized seizure, and non-seizure), as well as the mean adjacency matrices for PSG signals grouped by sleep stages. We observe that the learned graphs reflect the underlying seizure classes and sleep stages. We added qualitative explanations of the adjacency matrices in the last subsection “Visualization of graphs” in Section 5.2.
>
> Moreover, we visualize the adjacency matrix for traffic forecasting in Appendix F and Figure 5, and compare it to the adjacency matrix predefined based on sensor locations in Li et al. (2018). We find that the learned adjacency matrix (Figure 5a) does not exhibit block structures as in the predefined adjacency matrix (Figure 5b), which could be the reason why learning the traffic sensor graph results in lower performance than the predefined sensor graph (Table 4). This suggests that future work should explore integrating prior knowledge of the graph in the GSL layer.
>
> We acknowledge that the general audience may not be able to understand EEG signals, we moved the original Figure 2 (example dynamic EEG graphs and signals) to Appendix G and Figure 6. The average node degree in Figure 6a is 3.6, whereas the average node degree in Figure 6f is 4.2. Hence, the graph in Figure 6f is much denser than Figure 6a, which is expected given that the seizure spreads to all areas of the brain in the corresponding EEG signals in Figure 6f.
>
> **References**
>
> * Li, Y., Yu, R., Shahabi, C., & Liu, Y. (2018). Diffusion Convolutional Recurrent Neural Network: Data-Driven Traffic Forecasting. International Conference on Learning Representations. https://openreview.net/forum?id=SJiHXGWAZ

---

### Official Review · Reviewer_dUxW · 2022-10-24

**Confidence:** 4
**Correctness:** 3
**Technical Novelty And Significance:** 3
**Empirical Novelty And Significance:** 3
**Recommendation:** 5

**Clarity, Quality, Novelty And Reproducibility:**

I am stated many weaknesses that have to be revised yet, if these things are corrected, I stand quite positive about the general quality and clarity of the work.

For me, originality is in the proposed methodological insights, and the fact that the global pipleline is tested in three different situations (with comparison to SOTA) which are relevant, difficult and interesting.

The work appears to be well reproducicle, with a code well documented (even if I had no time to run it actually, only read it through).

**Strength And Weaknesses:**

Strength

1- the global architecture is sound with some novelties in the combination of elements.

2- numerical experiments are well conducted, on relevant and diverse tasks and datasets

3- even if there is no new theoretical content, there are many interesting methodological insights in Sections 3 and 4 explaining the rationales for the proposed method.

4- Excepted as mentioned below (about 4.4 and lack of details on the full pipeline), the article is well written and clear.

Weaknesses

1- The work combines S4 model (from Gu et al, 2022) and ideas of Graph Structural Learning (present in many works in the literature of these past years), yet it does not really offer a major new insight. For that, it is more an incremental work.

2- The work is not precise enough in the description of the exact model / architecture proposed. Section 4.4 sends back ton Figure 1 yet the figure is not, for me, evocative enough. I would prefer to have a details model (with clearly precised input and trainable parameters) in 4.4.

3- It seems that there is some limitation in that the work assumes that the graph comes necessarily from the multivariate series X, while in many settings data consists of separate graphs and multivariate series. And it is a question in itself to fathom or asses if the graph is well related to X. Here, it seems that one assumes to whe have a pipeline going from $X$ to $h$ (embedding from S4) then $W$ (learned graph). Maybe I didn't catch one point in eq. (8): is the learning of $E$ a way to incoporate pre-existing information on $G$ ? Or is $G$ always unknown and to be derived from $X$ ?

4- There is not that much variations or explorations of the parameters.

Some (more minor) points to improve are:

5- The S4 models could be presented with either more insight (how to choose $L$ ? Where goes $D*$ in (2) and  (3) ?), or taken as given by the original work of Gu and al. yet with the explanation that the model of eq. (4) has a rationale coming from state  space model of time series. The current presentation is too much in-between.

6- The pipeline from $X$ to the GNN layers should be clarified

7-  Some of the dimensions are not clear. For instance, are the T time slots cut in intervals of size r ? Or is there some superposition in sliding windows ?

8- As questioned above: is it opossible to have the situation where the input is $X$ and $G$ ?

9- For eq. (9), please use a different notation for $W$ on the left and $W$ on the right. (Also, the weights in eq (7) could use another letter than W... or Use A for the learned weighted adjacency matrix if you prefer to keep W for weights).

10- The effect of GSLk, as discussed in 5.2 on page 8 (and Table 4) doesn't appear to be major. Why that ? For DOD-H, why not simply use as baseline the K-NN graphs from the S4 embedding (without learning another graph) as comparison. This ablation study, remplacing GSP by pre-defined graph, looks to reduce the significance of the work so more comments and discussion is expected.

11- the graphs of figure 2 are hard to interpret because it is difficult, from the plotted time-series, to assess the state of the seizure in each clip. As it stands currently, it would be clearer to compare the obtained graphs to what is obtained in other studies (and remove the time series to save space).



**Summary Of The Paper:**

The paper combines the Structured State Space Model (S4) (to learn time behaviors of multivariate series) and Graph Structual Learning (by coding as an attention layer in inductive settings, and a learnable embedding for transductive settings, regularized with a k-NN graph and smoothness of (representation) of the series of the learnt graphs). This leads to a general proposed architecture which is used in three examples where one needs  to follow both some specific time dynamics (encoded in S4) and  a network structure (encoded in the proposed GSL).

**Summary Of The Review:**

My apppreciation is that the work has good potentials yet that there are several things that the authors should revise or anwer to.
My recommendation is that, as it stands, the work is "marginally below the acceptance threshold" but I would be happy to engage discussion and see improvement in the presentation of the work to have a more positive evaluation.

---

> ### Author Response · Authors · 2022-11-17
> **Responses to Reviewer dUxW (Part 1)**
>
> We thank the reviewer for the interest in our study and the detailed comments. Responses to specific comments are shown below.
>
> 1. *"The work combines S4 model (from Gu et al, 2022) and ideas of Graph Structural Learning (present in many works in the literature of these past years), yet it does not really offer a major new insight. For that, it is more an incremental work."*
>
> While the idea of graph structure learning exists, the idea of capturing dynamically evolving graph structures in time series has been largely overlooked in the literature, which is also supported by Reviewer aQfe that “capturing and utilizing dynamic graph structures over time is an important yet largely overlooked topic in graph-based representation learning”. Specifically, our novel contributions are (1) proposing a graph structure learning (GSL) method that learns dynamically evolving graph structures of multivariate signals, (2) developing a general GNN architecture leveraging Structured State Space model and GSL for graph-based spatiotemporal modeling of multivariate signals, (3) empirically showing that our proposed method consistently outperforms previous SOTA on three diverse data types and tasks, and that our GSL method learns meaningful graph structures. All of these contributions are new to the GNN and time-series community.
>
> 2. *"The work is not precise enough in the description of the exact model / architecture proposed. Section 4.4 sends back to Figure 1 yet the figure is not, for me, evocative enough. I would prefer to have a detailed model (with clearly precised input and trainable parameters) in 4.4."*
>
> We apologize for the confusion. We have now added the pseudocode for GraphS4mer in Algorithm 1 in Appendix A, which shows details of GraphS4mer. We have also updated Section 4.4 to make it clear the inputs of each component in GraphS4mer.
>
> 3. *"It seems that there is some limitation in that the work assumes that the graph comes necessarily from the multivariate series X, while in many settings data consists of separate graphs and multivariate series. And it is a question in itself to fathom or asses if the graph is well related to X. Here, it seems that one assumes to whe have a pipeline going from X to h (embedding from S4) then W (learned graph). Maybe I didn't catch one point in eq. (8): is the learning of E a way to incoporate pre-existing information on G? Or is G always unknown and to be derived from X?"*
>
> The reviewer is right that our full GraphS4mer architecture learns graph adjacency matrix $\mathbf{W}$ from the multivariate signals $\mathbf{X}$ in inductive settings (Equation 5). However, when the graph is known (can be defined from $\mathbf{X}$ or other prior knowledge), one can simply skip the GSL layer in GraphS4mer and use the predefined graph (together with S4 embeddings as node features) as inputs to the GNN layers. In fact, in Table 4, the experimental results for “GraphS4mer w/ predefined graph” were obtained using predefined graphs. For seizure detection from EEGs, we used the distance-based graph defined in Tang et al. (2022), which was constructed based on the physical distances between EEG sensors in the standard 10-20 EEG placement. For traffic forecasting, we used the traffic sensor graph defined in Li et al. (2018), which was constructed based on the physical locations of the traffic sensors. Both of these two predefined graphs were derived from prior knowledge of the sensor locations, rather than from $\mathbf{X}$.
>
> Equation 6 (originally, Equation 8) is designed for graph structure learning in transductive settings, where all multivariate signals over a short time interval share the same graph structure. One particular example is traffic forecasting, where all the traffic sensors form a graph, and the task is to forecast the future traffic speed for each sensor given the past traffic speed. In transductive settings, we use a randomly initialized, trainable embedding $\mathbf{E}^{(t)}$ to learn the adjacency matrix, which results in a universal graph structure within each time interval.  $\mathbf{E}^{(t)}$ is trained together with other model parameters for the specific task, and thus can learn the adjacency matrix that is optimized for the task. Currently, the learning of $\mathbf{E}^{(t)}$ does not incorporate pre-existing information about the graph. However, future work can explore integrating prior knowledge into GSL, as we have indicated in the "Conclusion and Future Work" section.
>
> We apologize for the typo that there should be a superscript (t) for $\mathbf{E}$ indicating that the learnable embedding is unique within each resolution interval. In the results reported in Table 3, resolution is chosen the same as the sequence length (i.e., 12 time steps), and thus the superscript (t) can be ignored.

---

> ### Author Response · Authors · 2022-11-17
> **Responses to Reviewer dUxW (Part 2)**
>
> 4. *"There is not that much variations or explorations of the parameters."*
>
> For hyperparameters, we tuned them using existing hyperparameter search tools, and the best set of hyperparameters were selected based on the best model performance on the validation set (see Appendix C “Details of model training procedures and hyperparameters”).
>
> We additionally examined the effect of temporal resolution $r$ on model performance. We now show the performance versus different values of $r$ in Figure 3 in Appendix D and subsection “Effect of temporal resolution” in Section 5.2. We observe that for inductive tasks (i.e., seizure detection and sleep staging), smaller value of $r$ tends to result in higher performance, which suggests that capturing dynamically varying graph structures is useful for inductive tasks. In contrast, for the transductive traffic forecasting task, larger value of $r$ tends to give better performance, which suggests that learning a static graph is sufficient for the traffic forecasting task.
>
> 5. *"The S4 models could be presented with either more insight (how to choose L? Where goes D∗in (2) and (3)?), or taken as given by the original work of Gu and al. yet with the explanation that the model of eq. (4) has a rationale coming from state space model of time series. The current presentation is too much in-between."*
>
> The S4 component in our GraphS4mer is largely taken as given by the original work and code implementation of Gu et al. (2022). However, we provide some background about S4 in Section 3.1 for readers before we formally introduce GraphS4mer. We have now shortened Section 3.1 to only include the background of SSM, its relation to convolution, and advantages of S4.
>
> The term $\mathbf{D} u(t)$ in the state space model (Equation 1) can be viewed as a skip connection of the input $u(t)$, and thus is omitted for exposition (Gu et al., 2022). $L$ is the sequence length, and it is set to the actual sequence lengths of the input multivariate signals in our experiments (e.g., $L$=12k for seizure detection from EEGs). We have added some clarifications about $\mathbf{D}$ and $L$ in Section 3.1.
>
> 6. *"The pipeline from X to the GNN layers should be clarified."*
>
> We apologize for the confusion. We have added the pseudocode for GraphS4mer in Algorithm 1 in Appendix A. The pseudocode describes in detail the pipeline from X to GNN layers.
>
> 7. *"Some of the dimensions are not clear. For instance, are the T time slots cut in intervals of size r ? Or is there some superposition in sliding windows?"*
>
> We denote $T$ as the sequence length (i.e., total number of time steps in the input signals). For convenience, we let the predefined resolution $r$ be an integer and assume that $T$ is divisible by $r$ without overlaps. We have clarified these points throughout Section 4.3 as well as in Algorithm 1 in Appendix A.
>
> 8. *"As questioned above: is it possible to have the situation where the input is X and G?"*
>
> As described above, if the input is $\mathbf{X}$ and its graph structure is known as $\mathcal{G}$, we can simply skip the GSL layer in GraphS4mer and use the predefined graph $\mathcal{G}$ as inputs to the GNN layers (together with S4 embeddings as node features). In Table 4, the experimental results for “GraphS4mer w/ predefined graph” were obtained using predefined graphs.
>
> 9. *"For eq. (9), please use a different notation for W on the left and W on the right. (Also, the weights in eq (7) could use another letter than W... or Use A for the learned weighted adjacency matrix if you prefer to keep W for weights)."*
>
> We have changed to use $\overline{\mathbf{W}}^{(t)}$ for the learned adjacency matrix from Equations 5-6 (originally, Equations 7-8), and $\mathbf{W}^{(t)}$for the adjacency matrix after adding the KNN graph in Equation 7 (originally, Equation 9). In addition, we have changed the notations for weight matrices for $\mathbf{Q}$ & $\mathbf{K}$ to $\mathbf{M}^Q$ and $\mathbf{M}^K$ in Equation 5 (originally, Equation 7), respectively, to distinguish them from adjacency matrices.

---

> ### Author Response · Authors · 2022-11-17
> **Responses to Reviewer dUxW (Part 3)**
>
> 10. *"The effect of GSL, as discussed in 5.2 on page 8 (and Table 4) doesn't appear to be major. Why that ? For DOD-H, why not simply use as baseline the K-NN graphs from the S4 embedding (without learning another graph) as comparison. This ablation study, remplacing GSL by pre-defined graph, looks to reduce the significance of the work so more comments and discussion is expected."*
>
> While Table 4 shows that the effect of GSL may not be significant for seizure detection and traffic forecasting tasks, one major advantage of our GSL method is that it can be applied to applications where a predefined graph structure is not available (e.g., sleep staging in our study). As suggested by the reviewer, we have performed an additional ablation experiment using the KNN graph as the predefined graph for sleep staging on DOD-H dataset. As shown in Table 4 (middle column), replacing GSL by the predefined KNN graph achieved a macro-F1 score of 0.765, whereas GraphS4mer achieved a macro-F1 score of 0.818. This result suggests that KNN graph may be suboptimal for the sleep staging task, and that GSL is useful for dynamically learning the graph structures when a predefined graph is not available.
>
> 11. *"The graphs of figure 2 are hard to interpret because it is difficult, from the plotted time-series, to assess the state of the seizure in each clip. As it stands currently, it would be clearer to compare the obtained graphs to what is obtained in other studies (and remove the time series to save space)."*
>
> We acknowledge that the EEG signals may be difficult to understand for the general audience. Therefore, we have moved the original Figure 2 to Figure 6 in Appendix G. We also added a visualization of the distance-based graph in Tang et al. (2022) in Figure 6. Note that this distance-based graph is fixed and shared across all EEGs, whereas our graphs are dynamically varying over time and are unique for each EEG.
>
> In addition, we updated Figure 2 to show mean adjacency matrices for EEG and PSG signals, grouped by seizure classes and sleep stages. We observe that the learned graphs reflect the underlying seizure classes and sleep stages. We have added explanations of these adjacency matrices in the last paragraph of Section 5.2.
>
> Moreover, we show the learned adjacency matrix for traffic forecasting, as well as the predefined traffic sensor graph in Li et al. (2018) in Appendix F and Figure 5. We find that the learned adjacency matrix (Figure 5a) does not exhibit block structures as in the predefined adjacency matrix (Figure 5b), which could be the reason why learning the traffic sensor graph results in lower performance than the predefined sensor graph (Table 4). This also suggests that future work should explore integrating prior knowledge of the graph in the GSL layer.
>
> **References**
>
> * Gu, A., Goel, K., & Re, C. (2022). Efficiently Modeling Long Sequences with Structured State Spaces. International Conference on Learning Representations. https://openreview.net/forum?id=uYLFoz1vlAC
> * Li, Y., Yu, R., Shahabi, C., & Liu, Y. (2018). Diffusion Convolutional Recurrent Neural Network: Data-Driven Traffic Forecasting. International Conference on Learning Representations. https://openreview.net/forum?id=SJiHXGWAZ
> * Tang, S., Dunnmon, J., Saab, K. K., Zhang, X., Huang, Q., Dubost, F., Rubin, D., & Lee-Messer, C. (2022, April). Self-Supervised Graph Neural Networks for Improved Electroencephalographic Seizure Analysis. International Conference on Learning Representations. https://openreview.net/pdf?id=k9bx1EfHI_-

---

### Official Review · Reviewer_HRnf · 2022-10-25

**Confidence:** 4
**Correctness:** 3
**Technical Novelty And Significance:** 2
**Empirical Novelty And Significance:** 2
**Recommendation:** 5

**Clarity, Quality, Novelty And Reproducibility:**

The technical contribution is not significant. There is no surprise to combine GSL and GNN for multivariate time series forecasting. The writing is clear and easy to follow. The authors only provided the range of hyperparameter tuning, the specific setting for the reported results is not provided, which may hinder the reproducibility.

**Strength And Weaknesses:**

Strength:
1. The motivation is clear and straightforward.

Weaknesses:
1. This paper proposed to taking the output of S4 as the input of GSL to overcome the limitation of GSL on capturing long-range dependencies. Two concerns: (1) Is only taking the output of S4 as input of GSL enough? More interactions are expected between S4 and GSL to guarantee long-range structure learning. (2) As the authors claimed "long range", how the performance of GRAPHS4MER varies for different "ranges" are not clear. Does it robust? It should be discussed in the experiment.
2. In 4.3, the authors first explained why they learn a graph for a time interval, then they claimed "our GSL layer learns a unique graph for each data point rather than a universal graph for all data points." Interval or a data point? It is confusing.
3. The interval r and the threshold k are two important hyperparameters. But the authors did not discuss how the performance varies to different r and k in the experiment.

**Summary Of The Paper:**

This paper proposed a method for long-range spatial-temporal multivariate signal forecasting, by combining S4 and graph structure learning to automatically learn the dynamic graph structures.

**Summary Of The Review:**

While the motivation and writing are clear, some concept descriptions and experiment settings need improvement, and the technical contribution is not significant.

---

> ### Author Response · Authors · 2022-11-17
> **Responses to Reviewer HRnf (Part 1)**
>
> Thank you for the detailed comments. Our responses to specific comments are listed below.
>
> 1. *"This paper proposed to taking the output of S4 as the input of GSL to overcome the limitation of GSL on capturing long-range dependencies. Two concerns: (1) Is only taking the output of S4 as input of GSL enough? More interactions are expected between S4 and GSL to guarantee long-range structure learning."*
>
> We adopted our current approach because of the following reasons. First, as shown in the original S4 paper (Gu et al., 2022), S4 initializes the matrix A in the state space model as the HiPPO matrix (Gu et al., 2020). HiPPO matrix allows the state x(t) to memorize the input u(t) (Gu et al., 2020). At each time step, the output of S4 essentially summarizes the history of the input at all of the previous time steps. Therefore, using the output of S4 is enough to model the long-range temporal dependency in the input signal. Second, our proposed approach is a simple yet effective way to combine S4 and graph structure learning (GSL). We have empirically shown that our approach outperforms previous state-of-the-art on three diverse tasks.
>
> We agree with the reviewer that more interactions between S4 and GSL may be needed to further improve long-range structure learning in the future. We have included this as a future work in the "Conclusion and Future Work" section in our manuscript.
>
> 2. *"As the authors claimed "long range", how the performance of GRAPHS4MER varies for different "ranges" are not clear. Does it robust? It should be discussed in the experiment."*
>
> We chose the three tasks because they cover a wide range of sequence lengths. Specifically, the EEGs in the seizure detection experiment have sequence lengths of 12k time steps; the PSGs in the sleep staging experiment have sequence lengths of 7.5k time steps; and the traffic forecasting task has sequence length of 12 time steps (see Section 5.1 “Experimental Setup”). GraphS4mer outperforms previous state-of-the-art on these 3 datasets, suggesting that our model is able to handle temporal dependencies in sequences ranging from 12 to 12k time steps. We have added some discussion about this point in Section 5.2 (last paragraph on page 7).
>
> 3. *"In 4.3, the authors first explained why they learn a graph for a time interval, then they claimed "our GSL layer learns a unique graph for each data point rather than a universal graph for all data points." Interval or a data point? It is confusing."*
>
> We apologize for the confusion. We have rephrased the sentence to “in inductive settings, our GSL layer learns **a dynamic graph within a short time interval $r$ for each multivariate signal** rather than a universal, static graph over all time intervals for all multivariate signals.” (third last paragraph on page 5). Moreover, we have provided the pseudocode for GraphS4mer in Algorithm 1 in Appendix A, which shows details of our model.
>
> 4. *"The interval r and the threshold k are two important hyperparameters. But the authors did not discuss how the performance varies to different r and k in the experiment."*
>
> In our experiments, we treated the threshold $\kappa$ as a hyperparameter and tuned it together with other hyperparameters using the validation set (see Appendix C “Details of model training procedures and hyperparameters”).
>
> To examine the effect of temporal resolution $r$ on model performance, we now show the performance versus different values of temporal resolution in Figure 3 in Appendix D and subsection “Effect of temporal resolution” in Section 5.2. We observe that for inductive tasks (i.e., seizure detection and sleep staging), smaller value of $r$ tends to result in higher performance, which suggests that capturing dynamically varying graph structures is useful for inductive tasks. In contrast, for the transductive traffic forecasting task, larger value of $r$ tends to give better performance, which suggests that learning a static graph is sufficient for the traffic forecasting task.

---

> ### Author Response · Authors · 2022-11-17
> **Responses to Reviewer HRnf (Part 2)**
>
> 5. *"The technical contribution is not significant. There is no surprise to combine GSL and GNN for multivariate time series forecasting."*
>
> We would like to emphasize that our proposed method is general and can be applied beyond time series forecasting, such as graph-level classification (e.g., seizure detection and sleep staging in our manuscript), graph-level regression, and node-level classification. In addition, we leverage the advantage of Structured State Space model (S4) to capture long-range temporal dependencies in multivariate signals. Specifically, we have made significant technical contributions by (1) proposing a graph structure learning (GSL) method that learns dynamically evolving graph structures of multivariate signals, (2) developing a general GNN architecture leveraging Structured State Space model and GSL for graph-based spatiotemporal modeling of multivariate signals, (3) empirically showing that our proposed method consistently outperforms previous SOTA on three diverse data types and tasks, and that our GSL method learns meaningful graph structures. All of these contributions are new to the GNN and time-series community, as supported by Reviewer aQfe that “capturing and utilizing dynamic graph structures over time is an important yet largely overlooked topic in graph-based representation learning” and by Reviewer dUxW that “originality is in the proposed methodological insights, and the fact that the global pipeline is tested in three different situations which are relevant, difficult and interesting”.
>
> 6. *"The authors only provided the range of hyperparameter tuning, the specific setting for the reported results is not provided, which may hinder the reproducibility."*
>
> We have provided the specific setting for the reported results in Appendix C “Details of model training procedures and hyperparameters” in our initial submission (was Appendix B in initial submission). Because each experiment had a different set of hyperparameters, we listed the specific hyperparameters in each subsection. For example, you can find the detailed hyperparameter settings for the seizure detection experiment under subsection “Model training and hyperparameters for seizure detection on TUSZ dataset” in Appendix C.
>
> **References**
>
> * Gu, A., Goel, K., & Re, C. (2022). Efficiently Modeling Long Sequences with Structured State Spaces. International Conference on Learning Representations. https://openreview.net/forum?id=uYLFoz1vlAC
> * Gu, Dao, Ermon, & Rudra. (2020). Hippo: Recurrent memory with optimal polynomial projections. Advances in Neural Information Processing Systems. https://proceedings.neurips.cc/paper/2020/hash/102f0bb6efb3a6128a3c750dd16729be-Abstract.html

---

### Official Review · Reviewer_aQfe · 2022-10-30

**Confidence:** 4
**Correctness:** 3
**Technical Novelty And Significance:** 3
**Empirical Novelty And Significance:** Not applicable
**Recommendation:** 3

**Clarity, Quality, Novelty And Reproducibility:**

Clarity: there are major points in the manuscript that need further improvement of clarity or providing more details, see my comments above.

Quality: overall quality of the manuscript is acceptable, with extensive experiments and ablation studies to support the author’s claims.

Reproducibility: the manuscript did not provide any source code, even though the datasets used for the experiments are public. Combined with the clarity issue in its methodology section, the proposed model and the experiments would be very difficult to reproduce.


**Strength And Weaknesses:**

Strength: Capturing and utilizing dynamic graph structures over time is an important yet largely overlooked topic in graph-based representation learning. Model performance is also quite good.

Weakness: Most importantly, the author did not provide sufficient details for the reader to understand how “Dynamic Graph Structure Learning” is performed (section 4.3). Specifically, the author mentioned that “each signal forms a graph,” which is confusing: does the statement indicate each variate forms a single graph; or does the multi-variate signal at a specific time point/short resolution form a single graph? If it is the latter case, it is unclear how the adjacency matrix W is learned. Furthermore, it is unclear how the KNN graph is constructed, especially if the length of the “short resolution” is small or, in the extreme case, a single time point. Secondly, it is unclear how the singles on multiple graphs W^(1)…W^(nd) were fed to the GNN layer. Based on the current description in 4.4 and Figure 1, it seems that the GNN layer is accepting signals defined on a single graph rather than multiple graphs. In addition, the ablation study performance shown in Table 4 indicates only a marginal benefit of using the dynamic graph structure.


**Summary Of The Paper:**

The manuscript proposed a time series embedding through structured state spaces + dynamic graph structure learning + GNN model to learn the representations of multi-variate time series data. The model was evaluated on three datasets (two graph-level classification tasks and one node-level forecasting task). Experiment results indicate superior performance of the proposed model over commonly-applied methods.

**Summary Of The Review:**

An interesting model for multivariate time series analysis by leveraging dynamic graph structure yet lacks sufficient algorithm details to understand the reproduce.

---

> ### Author Response · Authors · 2022-11-17
> **Responses to Reviewer aQfe**
>
> We thank the reviewer for the detailed comments. Responses to specific comments are listed below.
>
> 1. *"Most importantly, the author did not provide sufficient details for the reader to understand how “Dynamic Graph Structure Learning” is performed (section 4.3). Specifically, the author mentioned that “each signal forms a graph,” which is confusing: does the statement indicate each variate forms a single graph; or does the multi-variate signal at a specific time point/short resolution form a single graph? If it is the latter case, it is unclear how the adjacency matrix W is learned."*
>
> We apologize for the confusion. We propose different approaches for graph structure learning (GSL) for inductive and transductive settings.
>
> In inductive settings where the task is to predict on unseen multivariate signals (e.g., seizure detection from EEG signals and sleep staging from PSG signals), a graph is dynamically learned over a short time interval for each multivariate signal. This is the latter case stated by the reviewer. Over this interval, the adjacency matrix $W^{(t)}$ is learned according to Equation 5.
>
> Whereas in transductive tasks where all sensors form a graph (e.g., traffic forecasting), a graph is learned over a short time interval for all multivariate signals. Over this interval, the adjacency matrix $W^{(t)}$ is learned according to Equation 6.
>
> We have rephrased the 2nd and 3rd paragraphs of Section 4.3 to clarify the confusion. In addition, we have added the pseudocode for GraphS4mer in Algorithm 1 in Appendix A, which shows in detail how the adjacency matrices are learned.
>
> 2. *"Furthermore, it is unclear how the KNN graph is constructed, especially if the length of the “short resolution” is small or, in the extreme case, a single time point."*
>
> Similar to how dynamic graphs are learned in the inductive setting in Equation 5, the KNN graph is constructed based on the cosine similarity between nodes using the mean-pooled S4 embeddings within time ((t-1) x r, t x r], where t = 1, 2, …, T/r, T is the sequence length, and r is the pre-specified temporal resolution. We apologize for the typo — there should be a superscript (t) for $W_\text{KNN}$ to indicate that the KNN graph is varying over time. We have updated the second last paragraph on page 5 (Section 4.3) to clarify the construction of KNN graphs.
>
> 3. *"Secondly, it is unclear how the signals on multiple graphs W^(1)…W^(nd) were fed to the GNN layer. Based on the current description in 4.4 and Figure 1, it seems that the GNN layer is accepting signals defined on a single graph rather than multiple graphs."*
>
> The GNN layers take as inputs the S4 embeddings as node features, as well as the learned graphs $W^{(1)}$,..., $W^{(n_d)}$. This is now shown in the pseudocode in Algorithm 1 in Appendix A and 1st paragraph of Section 4.4.
>
> 4. *"In addition, the ablation study performance shown in Table 4 indicates only a marginal benefit of using the dynamic graph structure."*
>
> While Table 4 shows that the benefit of using the dynamic graph structure may be marginal for seizure detection, one major advantage of our graph structure learning (GSL) method is that it can be applied to applications where a predefined graph structure is not available (e.g., sleep staging task in our manuscript). As the DOD-H dataset for sleep staging has no predefined graph, we performed an additional experiment that replaced GSL with a KNN graph (constructed based on cosine similarity between nodes using S4 embeddings). As shown in Table 4 (middle column), replacing GSL with the KNN graph achieved a macro-F1 score of 0.765, whereas GraphS4mer achieved a macro-F1 score of 0.818. This suggests the effectiveness of using GSL, particularly on tasks where predefined graph structures are unavailable.
>
> 5. *"Reproducibility: the manuscript did not provide any source code, even though the datasets used for the experiments are public. Combined with the clarity issue in its methodology section, the proposed model and the experiments would be very difficult to reproduce."*
>
> We have submitted our source code as supplemental material in our initial submission. We also indicated the submission of our source code in “Reproducibility Statement” as follows: “Source code is included in supplemental materials.” We additionally added the sentence “Source code is included as supplemental material” in the last paragraph of Section 4.4.

---

### Author Response · Authors · 2022-11-17
**General responses to ACs & all reviewers**

Dear ACs and Reviewers,

We thank the reviewers for their close read of this manuscript and their insightful comments. Several important suggestions were made and we have considered each carefully and revised accordingly. Please find our replies to the reviewers below for detailed responses to the reviewers’ comments. We have also updated our manuscript with the following major changes that we believe have improved the paper:

1) Clarified details about graph structure learning and overall model architecture in Section 4.3 and 4.4. Moreover, we added the pseudocode for GraphS4mer in Algorithm 1 in Appendix A, which shows the details of GraphS4mer (all reviewers).

2) Added an additional experiment using a KNN graph as the predefined graph to replace graph structure learning (GSL) for the sleep staging experiment in Table 4. We found that GraphS4mer with GSL significantly outperforms that with KNN graph, suggesting the effectiveness of our GSL method (Reviewer dUxW).

3) Performed an additional interpretability analysis of learned graphs. In the updated Figure 2, we now show the mean adjacency matrices of EEG and PSG signals, grouped by seizure classes and sleep stages. Our interpretability analysis suggests that the learned graphs are meaningful and reflect the underlying seizure classes and sleep stages. In addition, we visualize the learned traffic sensor graph in Figure 5 in Appendix F. Moreover, we included an additional contribution of our work, “our GSL method learns meaningful graph structures that reflect the underlying seizure classes and sleep stages” in the Introduction & Conclusion and Future Work sections (Reviewers dUxW and FDcr).

4) Added results showing the effect of temporal resolution on model performance in subsection “Effect of temporal resolution” in Section 5.2 and Figure 3 in Appendix D (Reviewers HRnf and dUxW).

Sincerely,

Authors of paper 5618

---

> ### Comment · Reviewer_dUxW · 2022-11-18
> **acknowledgement of authors' reponses**
>
> I acknowledge to have read the answers and revisions to the numerous questions raised. The presentation of the method and results are now clarified and this is a good point. We will have to discuss to see if the originality of the work is substantial enough or not.
>
> Sincerely

---

### Decision · Program_Chairs · 2023-01-20

**Decision:**

Reject

**Justification For Why Not Higher Score:**

A major concern of the paper is on the innovation, as S4 model and dynamic graph modeling are already established in the community. Although the paper provides some improved results on benchmark datasets, the technical contribution is rather limited. The paper tackles long-range dependency problems in time series modeling, but it lacks insightful analysis and in-depth discussions on how long-range are characterized for different time series data, and the analysis on the interaction between long-range modeling and graph-based learning.

**Justification For Why Not Lower Score:**

N/A

**Metareview: Summary, Strengths And Weaknesses:**

This paper proposes GRAPHS4MER, a spatiotemporal model that combines S4 with graph structure learning to model long-range time series data. The model is tested on three benchmark datasets to show better performance than the state-of-the-art models.

The task of modeling long-range spatiotemporal dependency is interesting, difficult and important for time series analysis. The empirical results on three  benchmark datasets are supportive for the effectiveness of the proposed method.

However, as all the reviewers pointed out, the proposed method is a combination of the previous S4 model and a graph-based learning technique, which is rather an incremental contribution. The model design and experiment do not provide sufficient new insights for state-space modeling and graph-based learning. The long-range modeling has been provided by the S4 model, which can't be counted as a major contribution of this paper. Key questions such as how to quantify "long range" for different types of data and how much the long-range modeling can affect graph-based learning are not discussed (reviewer HRnf). Also, the discussion of existing works on dynamic graph modeling is not enough (reviewer FDcr). Other concerns include insufficient discussion on how to combine GSL with an existing (reviewer dUxW).

Although the reviewer improved the presentation of the paper and addressed some of the reviewer's questions, the major concerns still remain. Therefore, we recommend rejection.


**Summary Of Ac-Reviewer Meeting:**

This is not a borderline paper.